# Evolving Treatment Landscape of Frontline Therapy for Metastatic Urothelial Carcinoma: Current Insights and Future Perspectives

**DOI:** 10.3390/cancers16234078

**Published:** 2024-12-05

**Authors:** Whi-An Kwon, Min-Kyung Lee

**Affiliations:** 1Department of Urology, Myongji Hospital, Hanyang University College of Medicine, Goyang-si 10475, Republic of Korea; 2Department of Internal Medicine, Myongji Hospital, Hanyang University College of Medicine, Goyang-si 10475, Republic of Korea

**Keywords:** antibody-drug conjugates, cisplatin-based chemotherapy, enfortumab vedotin, frontline therapy, immune checkpoint inhibitors, metastatic urothelial carcinoma, platinum-based chemotherapy

## Abstract

Metastatic urothelial carcinoma (mUC) has traditionally been treated with cisplatin-based chemotherapy, but nearly half of these patients cannot receive cisplatin owing to comorbidities. Although immune checkpoint inhibitors have become crucial alternatives, optimizing first-line therapies for mUC remains challenging. Recently, the combination of enfortumab vedotin and pembrolizumab has shown significantly improved survival and response rates in cisplatin-ineligible patients, marking a substantial shift in frontline treatment. This review evaluates the evolving landscape of mUC therapies, focusing on the clinical outcomes of innovative combination regimens.

## 1. Introduction

Significant advances have recently been made in the therapeutic landscape of metastatic urothelial carcinoma (mUC) (Figure 1) [1]. Bladder cancer (BC), the tenth most common malignancy worldwide, accounted for approximately 573,000 new cases and 213,000 deaths in 2020. Urothelial carcinoma (UC), the most prevalent histological subtype, particularly in the United States and Europe, presents as metastatic cancer in 5–10% of patients at diagnosis [2]. While approximately 75% of newly diagnosed cases are categorized as non-muscle-invasive BC, the remaining 25% are identified as muscle-invasive BC or metastatic disease. The prognosis of mUC is particularly dire, with >90% of those with metastatic disease succumbing to the illness within 5 years [3].

Historically, the treatment of mUC has relied heavily on cisplatin-based chemotherapy regimens such as gemcitabine–cisplatin (GC) and methotrexate, vinblastine, doxorubicin, and cisplatin (MVAC), often supplemented with granulocyte colony-stimulating factor (G-CSF) for primary prophylaxis. These regimens yield a median (m) overall survival (OS) of approximately 14 months; however, up to 50% of patients are ineligible for cisplatin owing to comorbidities, poor performance status, or renal impairment. For these patients, alternatives include carboplatin and gemcitabine, as well as immune checkpoint inhibitors (ICIs) [4].

There has been a transformative shift in the treatment of mUC with the introduction of ICIs, antibody–drug conjugates (ADCs), and targeted therapies [5]. ICIs, such as pembrolizumab and atezolizumab, have gained regulatory approval for use in patients ineligible for cisplatin-based therapy, especially those with tumors expressing programmed death-ligand 1 (PD-L1), or those unsuitable for any form of platinum-based chemotherapy (PBC). Notably, pembrolizumab has demonstrated a significant survival benefit compared with second-line chemotherapy in patients who progress after PBC [6].

A significant advancement has been the molecular characterization of mUC, leading to the approval of targeted therapies, such as erdafitinib (fibroblast growth factor receptor (FGFR) inhibitor), for patients with FGFR-altered tumors [7]. These developments have broadened therapeutic options, particularly for patients who have progressed after or are ineligible for traditional chemotherapy.

Another notable milestone is the approval of avelumab as a switch maintenance therapy following first-line (1L) chemotherapy. This strategy may extend OS in patients responsive to initial treatment [8]. Recently, the results of the phase III EV-302 trial, which investigated the combination of enfortumab vedotin (EV) and pembrolizumab as the initial treatment, have challenged the established platinum-based treatment paradigm in mUC. The efficacy of this combination was further validated when the EV-302 trial successfully met its co-primary endpoints of OS and progression-free survival (PFS), marking a pivotal shift in the standard treatment of mUC [9].

Nevertheless, mUC management remains complex and challenging. Determining the optimal therapy for each patient, identifying appropriate treatment sequences, and exploring potential synergies among therapeutic agents are critical areas of ongoing research [10]. Real-world data suggest that only 30–40% of patients receiving primary therapy for mUC proceed to second-line treatment, underscoring the need for effective 1L treatments and improved strategies for subsequent lines of therapy [11].

In this review, we have explored the current therapeutic landscape of mUC, along with the challenges associated with optimizing treatment strategies and establishing effective treatment sequences, and the importance of personalized approaches in improving the survival outcome.

## 2. Types and Mechanisms of Drug Classes Used in the Frontline Treatment of mUC

### 2.1. Platinum-Based Chemotherapy

Platinum-based chemotherapy, such as cisplatin, primarily exerts its anticancer effects by forming DNA cross-links that disrupt the DNA double helix, thereby inhibiting critical processes such as DNA replication and transcription. This damage activates cellular stress responses, including DNA damage response pathways, leading to cell cycle arrest, and ultimately, apoptosis. Apoptosis is further enhanced by the activation of the intrinsic mitochondrial pathway and the tumor suppressor protein p53. Additionally, platinum compounds induce the production of reactive oxygen species, which further damage cellular components and amplify apoptosis. Although highly effective, platinum compounds also affect normal cells, leading to nephrotoxicity and neurotoxicity [12,13,14]. The multifaceted action of platinum-based drugs underscores their effectiveness in cancer treatment, particularly in BC; however, their efficacy needs to be balanced with adverse events (AEs).

### 2.2. ICIs

ICIs enhance the immune system’s ability to recognize and eliminate tumor cells [15] by blocking specific immune checkpoint pathways that normally maintain self-tolerance and modulate immune responses [16]. Under physiological conditions, immune checkpoints are crucial for preventing autoimmunity and limiting immune responses to avoid excessive tissue damage. However, tumors often exploit these pathways to evade immune surveillance [17].

Two of the most well-studied immune checkpoints targeted by ICIs are cytotoxic T lymphocyte-associated protein 4 (CTLA-4) and programmed cell death protein 1 (PD-1), along with its ligand, PD-L1. CTLA-4 is expressed on activated T cells and functions as an inhibitory receptor by competing with the co-stimulatory receptor CD28 for binding to B7 molecules (B7-1/CD80 and B7-2/CD86) on antigen-presenting cells. The binding of CTLA-4 to B7 molecules transmits an inhibitory signal that reduces T cell activation and proliferation. By blocking the interaction between CTLA-4 and B7, CTLA-4 inhibitors, such as ipilimumab, enhance T cell activation, thereby promoting a robust immune response against cancer cells [18,19,20].

The PD-1/PD-L1 pathway plays a similar role in regulating immune responses. PD-1 is expressed on T, B, and natural killer cells. When PD-1 binds to its ligands PD-L1 or PD-L2, it delivers an inhibitory signal that dampens T cell activity, proliferation, and cytokine production. Tumor cells often upregulate PD-L1 to suppress T cell-mediated immune responses, effectively protecting themselves from immune attack. ICIs targeting this pathway, such as PD-1 inhibitors (nivolumab, pembrolizumab, etc.) or PD-L1 inhibitors (atezolizumab, durvalumab, etc.), block the interaction between PD-1 and its ligands, thereby maintaining T cell activity.

The therapeutic efficacy of ICIs is a consequence of their ability to “release the brakes” on the immune system, allowing for enhanced T cell activation and sustained anti-tumor responses. This mechanism is effective across various malignancies, including melanoma, non-small cell lung cancer, renal cell carcinoma, and UC [21,22,23].

### 2.3. ADCs

ADCs are a class of targeted cancer therapies that combine the specificity of monoclonal antibodies with the potent cytotoxic effects of chemotherapeutic agents [24]. They are able to selectively target and kill cancer cells while minimizing damage to normal tissues [24,25]. The monoclonal antibody component of the ADC recognizes and binds to a specific antigen overexpressed on the cancer cell-surface, ensuring that the ADC preferentially targets tumor cells while sparing most normal cells, which typically show low or no expression of the antigen. Once an ADC binds to the target antigen on the cancer cell surface, the entire complex is internalized through receptor-mediated endocytosis. After internalization, the ADC is trafficked to the lysosome, where the antibody-linked cytotoxic drug is released through various mechanisms, depending on the type of linker used in the ADC design. Linkers may be cleavable, breaking down owing to the acidic environment of the lysosome or through enzymatic action, or non-cleavable, where the drug is released only after the entire ADC is degraded. Once the cytotoxic drug is released inside the cancer cell, it interferes with critical cellular processes. Common payloads used in ADCs include highly potent chemotherapeutic agents, such as microtubule inhibitors (auristatins, maytansinoids, etc.) or DNA-damaging agents (e.g., calicheamicin). Microtubule inhibitors disrupt the microtubule network within the cell, preventing cell division and leading to apoptosis. DNA-damaging agents induce breaks in the DNA strand, leading to cell death through apoptosis or mitotic catastrophe.

The high potential systemic toxicity of these drugs is mitigated by the selective delivery mechanism of the ADC, concentrating the drug’s effects within the tumor cells, thereby enhancing the treatment’s therapeutic index [26,27]. This targeted approach enables ADCs to deliver highly potent drugs directly to cancer cells, improving treatment efficacy while limiting off-target effects [28].

Figure 2 illustrates the mechanisms of action of the major drug classes used in the frontline treatment of mUC, including PBC, ICIs, and ADCs.

## 3. Patient Selection

Selecting the appropriate frontline therapy for patients with mUC is crucial for optimizing treatment outcomes. This process involves a comprehensive assessment of clinical, molecular, and patient-specific factors. Most patients with mUC are older, with the majority aged >65 years and approximately half aged >70 years [29]. This population often presents with comorbidities that complicate the use of standard cisplatin-based chemotherapy [30]. Consequently, many such patients are ineligible for cisplatin-based therapy, traditionally defined by factors such as an Eastern Cooperative Oncology Group Performance Status (ECOG PS) of ≥2 [31]. Given the nephrotoxic nature of cisplatin, renal function assessment is essential, with a creatinine clearance (CrCl) ≥ 60 mL/min typically serving as the threshold for cisplatin eligibility [32]. Patients with renal impairment, indicated by a CrCl < 60 mL/min, are often considered unsuitable for cisplatin and receive alternative treatments, such as carboplatin-based regimens or ICIs [33].

While the cisplatin ineligibility criteria are relatively well-defined, the ineligibility criteria for carboplatin-based chemotherapy remain unclear, leading to variability in clinical practice [34]. The approval of ICIs for 1L treatment of platinum-ineligible patients underscores the need for a consistent definition of platinum ineligibility [4]. Consensus definitions suggest that patients with an ECOG PS ≥ 3, CrCl < 30 mL/min, peripheral neuropathy grade > 3, or New York Heart Association heart failure class > 3 should be considered ineligible for PBC [35]. Additionally, patients with an ECOG PS of 2 and CrCl < 30 mL/min may also be ineligible for PBC. However, these criteria may not capture all relevant factors, such as advanced age, poorly controlled diabetes, or inadequate bone marrow reserve, which could further contraindicate PBC [4].

The first formal attempts to define cisplatin eligibility in mUC were made in the 1990s. A survey by the European Organization for Research and Treatment of Cancer identified a CrCl ≥ 60 mL/min and World Health Organization performance status of 0 or 1 as key requirements for cisplatin use [36]. These criteria were refined in 2011 by Galsky et al., whose guidelines for cisplatin ineligibility are widely used in clinical practice [37]. However, the exact renal function thresholds for cisplatin eligibility remain unclear, with CrCl cutoffs ranging from <45 mL/min to <60 mL/min, depending on the context [38].

Furthermore, molecular biomarkers have gained importance in guiding frontline therapy decisions, particularly regarding immunotherapy [39]. Patients with high PD-L1 expression, especially those ineligible for cisplatin, are more likely to benefit from ICIs, such as pembrolizumab or atezolizumab. However, PD-L1 expression should not be the sole factor guiding treatment decisions but rather part of a broader clinical assessment [40]. The histological subtype of UC also influences treatment decisions. Variants such as micropapillary, sarcomatoid, or neuroendocrine differentiation may respond differently to standard therapies, requiring a more personalized approach. In such cases, strategies such as neoadjuvant chemotherapy or enrollment in clinical trials may offer better outcomes [41]. Data on ICIs and new therapies for micropapillary, sarcomatoid, and neuroendocrine UC are limited. ICIs showed a 28% overall response rate (ORR) in 25 patients, and EV achieved a 35% ORR in 41 patients with micropapillary components [42,43]. Advanced sarcomatoid UC exhibits higher PD-L1 expression with 35–40% response rates to ICIs [42]; however, low nectin-4 expression may limit EV and pembrolizumab efficacy [44]. For neuroendocrine UC, single-agent anti-PD-1/PD-L1 inhibitors, as well as lurbinectedin and sacituzumab govitecan have demonstrated some efficacy in small studies [45,46,47,48], but the effectiveness of dual checkpoint blockade is uncertain [49,50], and low nectin-4 expression may limit EV’s benefits [43,44].

Patient preferences and quality of life are also critical in selecting frontline therapy. Having a thorough discussion with patients regarding the potential benefits and risks of each treatment option, specifically the impact on daily life and long-term prognosis, is essential [29]. This ensures that the chosen treatment aligns with the patient’s values and goals. Moreover, eligibility for clinical trials can provide access to novel therapies that may not be widely available [51].

Finally, the presence of specific genomic alterations, such as FGFR3 mutations, can guide the selection of targeted therapies, although these are currently used only in the second-line setting. Ongoing research may expand their use in frontline therapy, further personalizing the treatment approach for mUC [52].

Selecting frontline therapy for mUC requires a multifaceted evaluation, including performance status, renal function, comorbidities, molecular biomarkers, histological subtype, and patient preferences. A personalized approach is essential to optimize treatment outcomes of patients with mUC.

Table 1 provides an overview of the key factors involved in selecting the appropriate frontline therapy for patients with mUC.

## 4. Clinical Development

### 4.1. Chemotherapy

Cisplatin-based chemotherapy has long been the cornerstone of managing mUC, with an ORR of approximately 50%, PFS of up to 7 months, and mOS of 13–15 months [56,57]. Historically, the MVAC regimen was favored due to its superior OS compared to cisplatin monotherapy and other combinations like cisplatin, cyclophosphamide, and doxorubicin. However, application of MVAC has been limited by severe toxicities, including grade ≥ 3 leukopenia, neutropenic fever, mucositis, and gastrointestinal AEs, with a drug-associated mortality rate of 3–4% [57,58].

In a crucial phase III trial, the GC regimen showed long-term OS and PFS outcomes similar to those of MVAC but better tolerability regarding AEs and quality of life, leading many oncologists to prefer GC over MVAC [58]. Subsequently, high-dose-intensity chemotherapy with dose-dense (dd) MVAC plus G-CSF every 2 weeks demonstrated reduced toxicity and improved response rates compared to the conventional 4-week MVAC schedule [59,60]. Another phase III study reported similar outcomes between ddGC and ddMVAC, further supporting the use of either regimen based on patient tolerance [61].

For cisplatin-ineligible patients, carboplatin-based regimens have been employed. However, outcomes with carboplatin-based combinations are generally inferior to those with cisplatin-based regimens, although evidence is based on underpowered trials [62]. Recently, the DANUBE, IMvigor130, and JAVELIN-100 trials confirmed the superiority of cisplatin over carboplatin in platinum-eligible patients [63,64,65]. Split-dose administration of cisplatin has been considered for patients with borderline renal function, offering a potentially less nephrotoxic alternative while maintaining efficacy [66]. However, the VEFORA GETUG-AFU V06 study, comparing split-dose cisplatin with standard-dose carboplatin, was halted owing to excessive toxicities in the cisplatin arm [67].

Regarding chemotherapy cycles, consensus guidelines recommend 2–6 cycles of PBC [53]. Post hoc analysis from the JAVELIN Bladder 100 trial suggested that survival benefits from maintenance avelumab are consistent regardless of whether patients received 4, 5, or 6 cycles of chemotherapy [68]. Advances in treatment have diminished the role of second- and later-line chemotherapy regimens, which previously offered limited benefits over best supportive care (BSC) [69].

### 4.2. Immunotherapy

#### 4.2.1. 1L Monotherapy for Platinum-Ineligible Patients

ICIs such as atezolizumab and pembrolizumab have been extensively evaluated as monotherapy for cisplatin-ineligible patients with mUC 4. The phase II IMvigor210 trial evaluated atezolizumab as a 1L therapy in 119 patients with locally advanced or metastatic (la/m) UC who were ineligible for cisplatin-based chemotherapy [70]. The trial reported an ORR of 23%, with 9% patients achieving a complete response (CR) (mOS, 15.9 months). The revised ORRs by PD-L1 subgroup were 28% (95% confidence interval [CI]: 14–47) in IC2/3, 24% (95% CI: 15–35) in IC1/2/3, 21% (95% CI: 10–35) in IC1, and 21% (95% CI: 9–36) in IC0. Atezolizumab demonstrated a manageable safety profile, with grade 3 or 4 treatment-related AEs occurring in 16% patients. However, in May 2018, the Food and Drug Administration (FDA) issued a safety alert after early data from the IMvigor130 trial, and another study, suggested decreased survival of patients receiving atezolizumab as 1L monotherapy compared to those receiving PBC [71]. Consequently, the FDA restricted the use of atezolizumab as a 1L treatment to patients who were either ineligible for cisplatin-based chemotherapy with high PD-L1 expression (≥5% of immune cells) or those ineligible for any PBC, regardless of PD-L1 status [71]. The IMvigor130 trial, a multicenter phase III study [64], did not find a significant OS advantage for atezolizumab monotherapy over chemotherapy alone. Consequently, the manufacturer voluntarily withdrew the 1L indication for atezolizumab in mUC in November 2022 [11]. Nevertheless, the National Comprehensive Cancer Network (NCCN) Panel continues to recommend atezolizumab as a 1L option for patients ineligible for cisplatin-containing chemotherapy with tumors expressing PD-L1 or those ineligible for any PBC regardless of the PD-L1 status (category 2B recommendation) [53].

Subsequently, pembrolizumab was evaluated as 1L monotherapy for cisplatin-ineligible patients in the phase II KEYNOTE-052 trial [72], where it demonstrated an ORR of 24%, with 5% patients achieving a CR. Long-term outcomes remained consistent, with an ORR of 28.6% and mOS of 11.3 months. In patients with a PD-L1 combined positive score (CPS) ≥ 10, the ORR was 47.3%, with 20.0% patients achieving CR; for those with a PD-L1 CPS < 10, the ORR was 20.3%. The mOS was 18.5 months (95% CI, 12.2–28.5 months) in the CPS ≥ 10 group and 9.7 months (95% CI, 7.6–11.5 months) in the CPS < 10 group. However, similar to atezolizumab, the FDA issued a safety alert for pembrolizumab in May 2018 based on early findings from the KEYNOTE-361 trial that suggested decreased survival of patients receiving pembrolizumab monotherapy compared to those receiving PBC [73]. Therefore, the FDA restricted the use of pembrolizumab as a 1L treatment to patients who were ineligible for cisplatin-containing chemotherapy and had high PD-L1 expression (CPS ≥ 10) or those ineligible for any PBC regardless of the PD-L1 status [6]. The indication was further limited to patients ineligible for any PBC, with the NCCN Panel continuing to recommend pembrolizumab as a 1L treatment for cisplatin-ineligible patients regardless of the PD-L1 status, and specifically endorsing its use for those ineligible for any PBC [53].

Overall, while both atezolizumab and pembrolizumab provide new therapeutic options for cisplatin-ineligible patients with mUC, their use as 1L monotherapies has been restricted owing to concerns over OS, particularly in patients with low PD-L1 expression. The NCCN’s continued endorsement of these therapies, albeit with specific limitations, underscores the importance of careful patient selection in the management of mUC.

#### 4.2.2. 1L Combination Therapy

Combinations of ICIs and chemotherapy have generally yielded less favorable outcomes than anticipated [74]. For instance, the phase III trials IMvigor130 and KEYNOTE-361 evaluated the efficacy of atezolizumab and pembrolizumab, respectively, either as monotherapies or in combination with PBC, compared to chemotherapy alone [64,75]. The KEYNOTE-361 trial showed negligible improvements in PFS or OS when pembrolizumab was combined with PBC compared to chemotherapy alone in patients with mUC. Similarly, the final survival analysis from the IMvigor130 study did not demonstrate a significant OS advantage for the combination of atezolizumab with platinum and gemcitabine over chemotherapy alone in patients with mUC [76]. Nevertheless, exploratory data suggest a potential benefit of atezolizumab monotherapy in 1L cisplatin-ineligible patients with high PD-L1 expression, although the results were not statistically significant. Interestingly, a potentially greater benefit was observed when atezolizumab was combined with cisplatin instead of carboplatin, suggesting the need to determine the optimal chemotherapy regimen.

The phase III DANUBE trial further explored ICI combinations by evaluating durvalumab, an anti-PD-L1 agent, either as a monotherapy or in combination with tremelimumab, an anti-CTLA-4 agent, against standard chemotherapy in patients with mUC [63]. Unfortunately, the trial failed to meet its co-primary endpoints, which included OS comparisons between durvalumab monotherapy and chemotherapy in patients with high PD-L1 expression, as well as that between the durvalumab–tremelimumab combination and chemotherapy in the overall population. However, the promising activity observed in the PD-L1-high population led to the revision of the NILE protocol, which now focuses on untreated patients with mUC and high PD-L1 expression. The results of this global phase III trial are highly anticipated and may redefine treatment approaches for this subset of patients [77].

Contrastingly, the multinational phase III CheckMate 901 study demonstrated the efficacy of adding nivolumab to GC chemotherapy in 608 patients with previously untreated or unresectable mUC [78]. In this study, 251 patients who received the nivolumab combination continued with maintenance nivolumab for up to 2 years. After a median follow-up of 33.6 months, the results showed that the nivolumab plus GC combination significantly improved mOS compared to GC alone (21.7 vs. 18.9 months; hazard ratio [HR], 0.78; 95% CI, 0.63–0.96; *p* = 0.02). Although the median PFS (mPFS) was similar between the two groups (7.9 vs. 7.6 months; *p* = 0.001), the PFS curves began to diverge over time, with a higher percentage of patients remaining progression-free at 12 months in the nivolumab combination group (34.2% vs. 21.8%). Furthermore, the ORR was notably higher in the nivolumab combination group (57.6% vs. 43.1%), with a CR rate of 21.7% compared to 11.8% in the chemotherapy-alone group. Among patients with tumor PD-L1 expression ≥1%, the nivolumab combination demonstrated a better HR compared to gemcitabine–cisplatin alone. The HR for OS and PFS as assessed by a central review were 0.75 (95% CI, 0.53–1.06) and 0.60 (95% CI, 0.41–0.81), respectively. However, the combination therapy was associated with a higher incidence of grade ≥ 3 treatment-related AEs (TRAEs), occurring in 61.8% of patients compared to 51.7% in the chemotherapy-alone group. These findings led to the regimen being designated as a category 1 recommendation by the NCCN Panel [53]. The results of the CheckMate-901 trial stand in contrast to those of earlier phase III trials, such as the KEYNOTE-361 and IMvigor130, which did not show significant improvements in either OS or PFS when ICIs such as pembrolizumab or atezolizumab were added to PBC for 1L treatment of mUC. Differences in the immunomodulatory effects of cisplatin and carboplatin may partially explain the discrepancies between these trials [78]. Exploratory analyses of both KEYNOTE-361 and IMvigor130 suggested that combining ICIs with cisplatin-based therapy, but not carboplatin-based therapy, resulted in better outcomes. Specifically, the outcomes were better when patients with pretreated tumors with higher PD-L1 expression were treated with GC than with gemcitabine–carboplatin [79]. Furthermore, single-cell RNA sequencing of circulating immune cells in the IMvigor130 trial revealed that GC, but not gemcitabine–carboplatin, upregulated immune-related transcriptional programs, including those involved in antigen presentation [80]. Therefore, cisplatin-based chemotherapy may have particularly favorable immunogenic effects when combined with ICIs in mUC treatment, as observed in the CheckMate-901 trial, highlighting the importance of chemotherapy selection for optimizing the efficacy of combination therapies in mUC.

#### 4.2.3. Maintenance Therapy

Avelumab as maintenance therapy following 1L PBC is considered a significant advancement in mUC treatment [81]. This regimen is based on findings from the phase III JAVELIN Bladder 100 trial 62, which demonstrated that avelumab significantly prolongs OS compared to BSC alone in patients who achieved either a response or stable disease after 1L chemotherapy. Specifically, the trial showed an mOS of 21.4 months for patients treated with avelumab, compared to 14.3 months for those who received BSC alone, corresponding to an HR of 0.69 (95% CI, 0.56–0.86; *p* = 0.001). This OS benefit was consistent across all prespecified subgroups, including patients with PD-L1-positive tumors.

In the PD-L1-positive population, the avelumab group had significantly longer OS compared to the control group, with a 1-year survival rate of 79.1% (95% CI, 72.1–84.5) versus 60.4% (95% CI, 52.0–67.7) in the control group (stratified HR, 0.56; 95% CI, 0.40–0.79; *p* < 0.001). Among patients with PD-L1-negative tumors, the mOS was 18.8 months (95% CI, 13.3–22.5) in the avelumab group compared to 13.7 months (95% CI, 10.8–17.8) in the control group (stratified HR, 0.85; 95% CI, 0.62–1.18). Additionally, avelumab was also associated with a longer PFS compared to BSC alone. Importantly, preplanned subgroup analyses demonstrated that the OS benefit of avelumab was significant regardless of previous treatment with cisplatin or carboplatin or the response to chemotherapy. Despite a higher incidence of subsequent treatments in the control group, including with ICIs, the 12-month OS was substantially higher in the avelumab arm (71%) compared to that in the BSC arm (58%). The incidence of grade ≥ 3 AEs was higher in the avelumab group (47.4%) compared to that in the BSC group (25.2%). Yet, the long-term safety profile of avelumab remained manageable, with the most common TRAEs being pruritus, hypothyroidism, fatigue, diarrhea, and asthenia. Based on the findings of the JAVELIN Bladder 100 trial, the NCCN assigned a category 1 recommendation to avelumab maintenance therapy for patients with mUC who do not experience disease progression after 1L PBC 65. Subsequent extended follow-up data, with a median duration of 38 months, confirmed the durability of the OS and PFS benefits of avelumab, with an OS of 23.8 months versus 15.0 months (HR 0.76; 95% CI, 0.631–0.915; *p* = 0.0036) and PFS of 5.5 months versus 2.1 months (HR 0.54; 95% CI, 0.457–0.645; *p* < 0.0001) [82].

Real-world studies conducted in Italy (READY study) and France (AVENANCE study) have further validated the clinical benefits and manageable safety profile of avelumab as a 1L maintenance therapy for patients with mUC [83,84]. These studies reported 12-month OS rates of 65.4–69.2% and mPFS of 5.7–8.1 months, consistent with the findings of the JAVELIN Bladder 100 trial.

Following the results of the CheckMate-901 trial, nivolumab has also emerged as a viable maintenance therapy option for patients with mUC who respond to nivolumab plus 1L cisplatin-based chemotherapy. The trial demonstrated that nivolumab plus GC significantly improved outcomes in these patients, leading the NCCN to recommend nivolumab as an alternative maintenance therapy alongside avelumab [53]. This expanded recommendation gives clinicians more flexibility when selecting an appropriate maintenance therapy based on patient-specific factors.

While both avelumab and nivolumab maintenance therapies have significantly improved the outcomes of patients with mUC, ongoing phase II and phase III trials are exploring combinations of these agents with other therapies as 1L maintenance treatments. The optimal second-line therapy for patients with disease progression during or after maintenance therapy with either avelumab or nivolumab is still under investigation, and enrollment in clinical trials is strongly recommended [85].

### 4.3. EV Combination Therapy

The combination of the ICI pembrolizumab with the ADC EV has been explored extensively for mUC treatment 10. Initial studies were conducted within certain phase Ib/II EV-103 cohorts, mainly focusing on cisplatin-ineligible patients with previously untreated la/mUC [86]. In cohort A, 45 patients received the combination therapy, resulting in an ORR of 73.3% and CR of 15.6%. However, the treatment was associated with significant AEs, including peripheral sensory neuropathy (55.6%), fatigue (51.1%), and alopecia (48.9%), with 64.4% of patients experiencing grade ≥ 3 AEs, and one treatment-related death. Cohort K of the EV-103 trial randomized cisplatin-ineligible patients to receive either EV alone or in combination with pembrolizumab. The combination therapy achieved a confirmed ORR of 64.5% (95% CI, 52.7–75.1%) compared to 45.2% (95% CI, 33.5–57.3%) for EV monotherapy. The median duration of response was not reached for the combination, whereas it was 13.2 months for the monotherapy.

The combination was further evaluated in the phase III EV-302 trial in 886 patients with previously untreated la/mUC [9]. Patients were randomized to receive either EV plus pembrolizumab or standard chemotherapy with gemcitabine in combination with either cisplatin or carboplatin. After a median follow-up of 17.2 months, the combination therapy significantly improved PFS and OS. Specifically, the mPFS was 12.5 months with the combination versus 6.3 months with chemotherapy (HR, 0.45; 95% CI, 0.38–0.54; *p* < 0.001), and the mOS was 31.5 months versus 16.1 months (HR, 0.47; 95% CI, 0.38–0.58; *p* < 0.001). Additionally, the confirmed ORR was 67.7% for the combination compared to 44.4% for chemotherapy, with CR in 29.1% of patients in the combination group versus 12.5% in the chemotherapy group. Notably, grade ≥ 3 TRAEs occurred in 55.9% of patients receiving the combination therapy, compared to 69.5% of those receiving chemotherapy.

The impressive outcomes from the EV-302 trial led to the combination of EV and pembrolizumab being recognized as the preferred 1L systemic therapy for patients with advanced UC or mUC, regardless of cisplatin eligibility. This regimen has received a category 1 designation from the NCCN panel, solidifying its status as a new standard of care in this setting [53].

#### EV-302 and JAVELIN Paradigm Versus CheckMate-901

The role of GC/nivo (gemcitabine/cisplatin with nivolumab) and the JAVELIN paradigm in the management of mUC is being reevaluated in light of the impressive results of EV/pembro (enfortumab vedotin with pembrolizumab) [9]. The usefulness of GC/nivo in cisplatin-eligible patients is because of several factors. Firstly, cisplatin-based combination therapies have historically cured 5–15% of patients [58,60]. The long-term CR observed in approximately 22% of patients receiving GC/nivo supports the possibility of achieving a cure in a subset of patients [78]. While EV/pembro may also lead to cure, long-term follow-up is required to confirm this potential. Secondly, compared to EV/pembro, GC/nivo offers a finite duration of chemotherapy, providing advantages of reduced toxicity and improved quality of life during treatment [9,78]. Additionally, GC/nivo is more cost-effective compared to EV/pembro, making it a financially viable option for many healthcare systems and patients [87,88]. Moreover, GC/nivo may be considered for patients with uncontrolled diabetes mellitus or liver dysfunction owing to the specific safety and tolerability considerations associated with EV [78,89,90].

The identification of predictive biomarkers for CR is crucial for implementing precision medicine in mUC. Disease confined to lymph nodes is a favorable prognostic factor for both cisplatin-based chemotherapy and PD-1 inhibition, suggesting the aggressive use of GC/nivo in this patient group [91,92]. Additionally, ERCC2 mutations, previously validated for predicting pathological CR with neoadjuvant cisplatin-based chemotherapy, may serve as biomarkers for predicting CR with cisplatin-based therapy [93].

Given that aggressive multi-agent combination therapies are feasible only in selected patients in real-world settings, all 1L regimens, including EV/pembro, GC/nivo, and the JAVELIN paradigm, have legitimate roles in clinical practice [53].

Despite the availability of maintenance immunotherapy, its implementation in real-world practice is limited. In the EV-302 study, only 32.2% patients received a PD-1/L1 inhibitor as maintenance therapy after PBC, and only 20% patients in the GC arm of the CheckMate-901 study received a PD-1/L1 inhibitor before progression. The suboptimal application of the JAVELIN paradigm may be attributed to attrition from disease progression, persistent toxicities, poor ECOG performance status (PS), frailty, comorbidities, and patient decisions [85]. Therefore, potent 1L regimens that provide early and durable benefits, such as EV/pembro and GC/nivo, both of which induce rapid responses within 2–3 months, are expected to replace the JAVELIN paradigm for most patients. Nonetheless, in frail or vulnerable cisplatin-ineligible patients (including potentially some with ECOG PS 2), employing the JAVELIN paradigm of carboplatin/gemcitabine followed by maintenance avelumab may be a safer option.

Table 2 summarizes the key clinical trials on frontline therapy for mUC, highlighting the interventions, outcomes, and AEs associated with each treatment strategy.

### 4.4. Real-World 1L Treatment Patterns and Survival Rates for la/mUC

Several retrospective studies have analyzed real-world 1L treatment patterns and OS in patients with la/mUC in the United States and Germany [95,96,97]. These regions were chosen owing to their high incidence of UC and availability of comprehensive healthcare data that support real-world analysis.

A large-scale analysis of Medicare fee-for-service claims from 2015 to 2022 involving 13,104 U.S. patients with UC revealed that PBC remains the most commonly administered 1L treatment. Specifically, 19.7% patients received cisplatin and 28.4% received carboplatin. ICI monotherapy was also prevalent, being administered to 34.8% of patients, particularly those with comorbidities such as renal disease (39.9%), chronic obstructive pulmonary disease (36.1%), and diabetes (32.2%). Despite cisplatin-treated patients showing the longest median OS at 17.0 months, overall outcomes were generally poor across all treatment groups, largely due to disease progression, limited efficacy of available treatments, and patient comorbidities that impacted tolerability and response [97].

Following the FDA approval of avelumab maintenance therapy in June 2020, a study utilized the U.S. Oncology Network electronic health records from December 2019 to September 2022 and assessed 1072 patients with la/mUC (median age: 73 years, 74.9% male). The 1L treatments included ICI monotherapy (43.8%), PBC alone (37.3%), and PBC followed by avelumab maintenance (10.4%). Notably, ICI monotherapy was frequently used even in patients eligible for PBC. Patients receiving PBC with subsequent avelumab maintenance exhibited the longest median OS (20 months for cisplatin plus avelumab and 18.4 months for carboplatin plus avelumab), supporting avelumab maintenance as a standard option for those without disease progression on PBC. Thus, incorporating maintenance therapy into clinical practice may potentially extend patient survival, especially in those responding well to initial PBC [96].

In Germany, the CONVINCE retrospective multicenter study evaluated 188 patients treated between 2019 and 2021 (median age: 70 years, 72.3% male). The majority (76.1%) received PBC, predominantly combined with gemcitabine, while 19.1% received ICI monotherapy, mainly atezolizumab or pembrolizumab. A small group received other non-PBC treatments. The study found high adherence to treatment guidelines recommending PBC as the 1L therapy. The emerging use of ICI therapies, including avelumab maintenance, indicates potential for improved patient outcomes and necessity for further real-world research to assess their integration into clinical practice [95].

Collectively, these studies demonstrate that while PBC remains the standard 1L treatment associated with longer OS in patients with la/mUC, the OS rates are suboptimal, primarily due to disease biology, treatment tolerability, and patient-specific factors such as comorbidities. The increasing adoption of ICI therapies and maintenance treatments such as avelumab may enhance outcomes. However, the substantial use of ICI monotherapy, even among patients who could tolerate PBC, suggests opportunities to optimize treatment strategies. These findings underscore an unmet need for more effective 1L therapies to improve survival of patients with la/mUC.

### 4.5. Survival Benefits of the Different Therapeutic Protocols

Among cisplatin-eligible patients, gemcitabine plus cisplatin, the historical standard, achieved a mOS of 13.8 months [57]. The addition of nivolumab (CheckMate 901) improved the OS to 21.7 months [78]. EV plus pembrolizumab (EV-302) reported the longest OS at 31.5 months [9].

For cisplatin-ineligible patients, gemcitabine plus carboplatin resulted in a median OS of 9.3 months [98], highlighting the limitations of carboplatin-based chemotherapy, which is generally less effective owing to its lower potency and reduced ability to form DNA cross-links compared to cisplatin. Pembrolizumab monotherapy (KEYNOTE-052), approved for platinum-ineligible patients, demonstrated an OS of 11.3 months but remains less effective than combination therapies [72].

Avelumab maintenance therapy (JAVELIN 100), following disease control with PBC, significantly prolonged survival [65]. The median OS was 25.8 months for patients receiving carboplatin during induction and 31.5 months for those receiving cisplatin. This highlights the specific role of avelumab in maintaining disease control and extending OS, particularly in patients who initially respond to platinum-based induction therapy.

However, outcomes for cisplatin-ineligible patients remain suboptimal, as reflected by the poorer OS with carboplatin-based or immunotherapy monotherapies, emphasizing the need for further research, including the development of novel targeted therapies and improved ADCs, and exploring combination regimens with better response rates.

## 5. Future Perspectives and Discussion

The evolving landscape of frontline therapy for mUC presents significant advancements, with many ongoing Phase III and II clinical trials actively contributing to the refinement of future treatment paradigms (Table 3). Recent developments have introduced promising therapeutic agents and combination therapies for mUC, such as EV with pembrolizumab, which may potentially enhance patient outcomes. However, challenges persist in determining the optimal sequencing, duration, and combination of these therapies, as well as in identifying predictive biomarkers to guide clinical decision-making.

### 5.1. Challenges in Optimizing EV Plus Pembrolizumab for Frontline mUC Treatment

The combination of EV with pembrolizumab in the frontline setting has shown promise as a potential standard of care for all patients, regardless of their eligibility for PBC. This raises the possibility of reserving chemotherapy and targeted agents for later lines of treatment, adding complexity to treatment decisions [85]. However, several questions remain, such as the optimal duration of therapy, sequencing of treatments, and potential for EV rechallenge after discontinuation owing to toxicity. Additionally, the role of EV plus pembrolizumab versus EV alone in patients who have previously received adjuvant immunotherapy is worth investigating [99,100]. Clinical trials are exploring these questions, including alternative dosing schedules and maintenance strategies, such as reducing the dosing frequency of EV or implementing maintenance pembrolizumab after an initial course of EV plus pembrolizumab [101].

As the mUC treatment landscape continues to evolve, the sequencing of therapies is becoming increasingly complex. The potential widespread adoption of EV plus pembrolizumab as frontline therapy may shift the use of chemotherapy and other targeted agents to subsequent lines, necessitating a re-evaluation of the current treatment paradigms, particularly for patients who experience disease progression after 1L therapy [9,102]. However, questions remain regarding the role of chemotherapy after progression on EV plus pembrolizumab. Ongoing trials are examining the efficacy of various sequences, such as gemcitabine–carboplatin following EV plus pembrolizumab, and combinations of newer agents such as sacituzumab govitecan with pembrolizumab [103,104].

### 5.2. Biomarkers of Response

Identifying predictive biomarkers is crucial for frontline mUC treatment to guide therapeutic decisions. Predicting which patients will benefit most from specific regimens, such as EV plus pembrolizumab, GC with nivolumab followed by maintenance, or other chemo-immunotherapy combinations, is essential for optimizing outcomes [105]. Considerable progress has been made in this area, particularly with biomarkers related to immunotherapy, such as PD-L1 expression, microsatellite instability, defective mismatch repair phenotype, and tumor mutational burden [106]. A recent meta-analysis of 14 studies showed that PD-L1-positive UC is associated with higher response rates and improved survival outcomes in patients treated with ICIs. Although PD-L1 positivity suggests a better prognosis, it is not a reliable predictive biomarker for ICI response [40].

Emerging biomarkers, including circulating tumor DNA (ctDNA) and gut microbiota, have shown promise but require further validation [107]. The STING trial underscored the feasibility and reliability of plasma ctDNA in identifying actionable targets and selecting genotype-specific therapies in mUC [108].

### 5.3. Managing Toxicities in Frontline Therapy

The emergence of EV plus pembrolizumab as a frontline treatment is associated with challenges in managing the associated toxicities. Peripheral neuropathy and skin reactions are among the most-reported AEs of EV, requiring careful management, especially in patients with comorbidities such as diabetes. Additionally, a recently reported case of lung toxicity emphasizes the need for close pulmonary monitoring during EV treatment [109]. While these toxicities are generally manageable, they can lead to treatment interruptions or discontinuation [9,110]. Given the increasing use of EV plus pembrolizumab, there is a growing focus on optimizing dosing schedules to minimize toxicity while maintaining efficacy. Ongoing clinical trials are exploring fixed-duration EV therapy followed by pembrolizumab maintenance as a potential approach to improve the quality of life of patients [111].

### 5.4. Combining Low-Risk Dietary and Metabolic Therapies with Standard Cancer Treatments to Enhance Efficacy and Mitigate Toxicity

The therapeutic effectiveness of standard cancer therapies, such as ICIs and ADCs, is limited by severe AEs. Therefore, there is a need for low risk combined therapies that enhance the immune system. Plant-based dietary nutrients, such as ginger derivatives, show anti-tumor effects and can mitigate ICI-related AEs [112]. High-dose ascorbic acid has significant potential but is underutilized [113,114,115]. Metabolic interventions such as fasting, avoiding sugar to starve tumor cells, and increasing tumor pH to inhibit growth have been considered [116]. Combining these measures with standard treatments has the potential to improve efficacy and reduce toxicity.

### 5.5. Clinical Significance of Newer Treatments Other than EV-302

The currently recommended treatment algorithm (Figure 3) for mUC outlines strategies based on cisplatin eligibility and patient-specific factors and incorporates recent evidence to guide therapeutic decisions. For patients with visceral and/or bone metastasis, upper tract UC, high-risk cardiac disease, or good performance status, regardless of cisplatin eligibility, EV/pembro is recommended because of improved survival outcomes. In cisplatin-eligible patients with comorbidities such as poorly controlled diabetes or retinal/corneal abnormalities, GC/nivo followed by maintenance nivolumab or GC followed by maintenance avelumab is advised, enabling a tailored approach based on disease response and tolerability. For cisplatin-ineligible patients with peripheral neuropathy (Grade ≥ 2) or other vulnerabilities, gemcitabine–carboplatin followed by maintenance avelumab remains the standard of care. Finally, for patients who are platinum-ineligible, pembrolizumab monotherapy is the primary option, particularly for those who cannot tolerate platinum-based regimens. This algorithm reflects a personalized approach, optimizing the use of combination therapies such as EV/pembro and maintenance strategies with avelumab or nivolumab, while addressing the unmet needs of patients with significant comorbidities or platinum ineligibility.

## 6. Conclusions

The evolving landscape of frontline mUC treatment is marked by a shift from traditional PBC to more innovative and targeted therapies, such as EV plus pembrolizumab and GC plus nivolumab followed by nivolumab maintenance. These regimens have significantly improved survival outcomes, offering effective 1L options for a broader range of patients, including those previously ineligible for cisplatin-based treatments. As EV plus pembrolizumab becomes increasingly favored owing to its substantial impact on PFS and OS, managing its associated toxicities, such as neuropathy and other AEs, remains a critical challenge. Similarly, the GC plus nivolumab combination, with its potential for durable responses, particularly in node-only disease, offers new hope for curative outcomes.

Future efforts should focus on optimizing treatment strategies, identifying biomarkers, and further understanding the drug resistance mechanisms. As new frontline therapies are integrated into clinical practice, their long-term impact on outcomes and quality of life of patients will continue to shape the future of mUC treatment.

## Figures and Tables

**Figure 1 cancers-16-04078-f001:**
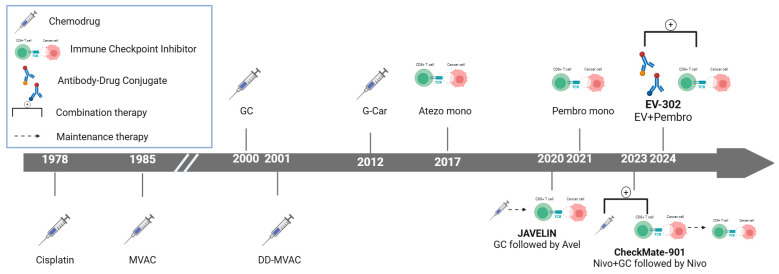
Evolution of frontline treatments for metastatic urothelial carcinoma. Since the approval of cisplatin in 1978, treatment options for urothelial cancer remained fairly static. While there were some modifications in chemotherapeutic regimens, such as dose-dense methotrexate, vinblastine, doxorubicin, and cisplatin (dd-MVAC), no fundamentally new treatment types emerged for decades. This situation changed in 2017 with the introduction of immune checkpoint inhibitors (ICIs), which marked a turning point in the management of metastatic urothelial carcinoma. Following the approval of the first ICIs, several new agents became available, including additional ICIs and antibody-drug conjugates. These developments were driven by pivotal phase III trials (JAVELIN, CheckMate-901, EV-302) that evaluated novel chemotherapies and other treatments. Atezo, atezolizumab; avel, avelumab; dd-MVAC, dose-dense methotrexate, vinblastine, doxorubicin and cisplatin; EV, enfortumab vedotin; GC, gemcitabine, cisplatin; G-Car, gemcitabine, carboplatin; nivo, nivolumab; pembro, pembrolizumab; mono, monotherapy.

**Figure 2 cancers-16-04078-f002:**
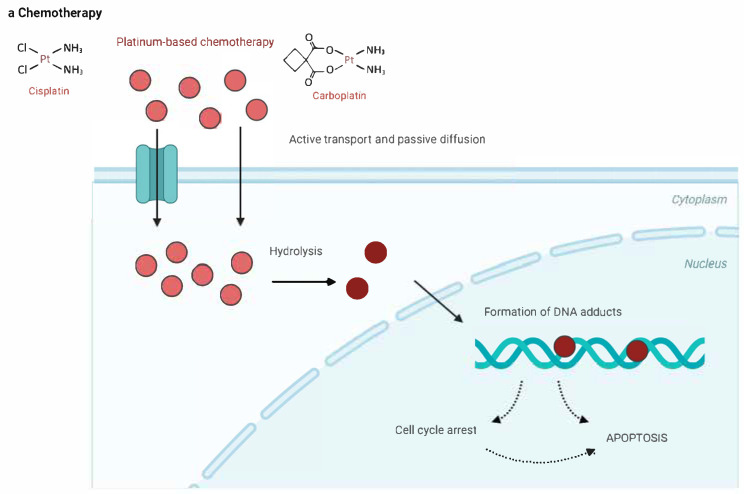
Drug classes and their mechanisms in the frontline treatment of metastatic urothelial carcinoma. (**a**) Platinum-based chemotherapy works by disrupting cell division. Platinum compounds bind to the DNA of cancer cells, creating cross-links that impede normal DNA replication and cell division. This DNA damage hampers the cancer cells’ growth and spread. (**b**) Immuno-therapy boosts the patient’s immune system to fight against cancer cells. It works by blocking immune checkpoint pathways, such as PD-1, PD-L1, and CTLA-4, which tumor cells use to evade immune detection. By inhibiting these pathways, immunotherapy enables the immune system to attack and eliminate cancer cells. (**c**) Antibody–drug conjugates (ADCs) consist of a cancer-specific antibody linked to a cytotoxic agent. Once the ADC attaches to the cancer cells, it is absorbed, and the toxic drug is released inside the cell, leading to cancer cell death. Enfortumab vedotin targets nectin-4, which is commonly overexpressed in urothelial cancer cells, while sacituzumab govitecan targets tumor cells through the anti-Trop-2 antibody. APC, antigen-presenting cell; CTLA-4, cyto-toxic T-lymphocyte associated protein 4; MMAE, monomethyl auristatin; PD-1, programmed cell death protein 1; PD-L1, programmed death-ligand 1; PD-L2, programmed death-ligand 2; SN-38, 7-ethyl-10-hydroxycamptothecin.

**Figure 3 cancers-16-04078-f003:**
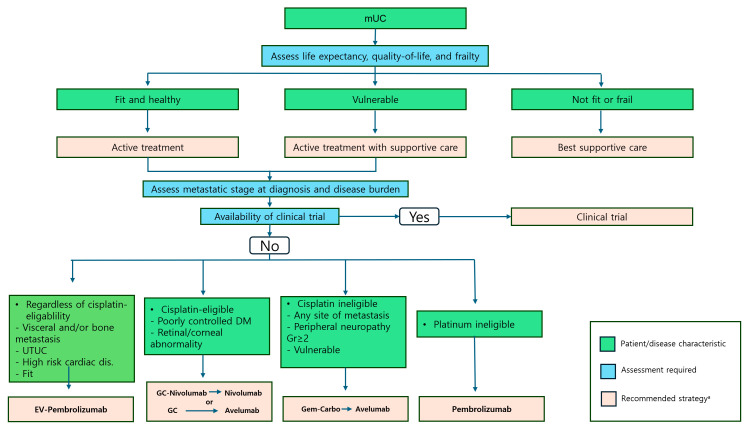
Recommended treatment approaches for frontline patients with mUC. All treatment decisions should be made only after thoroughly discussing the benefits and risks with the patient and/or their caregiver. Abbreviations: DM, diabetes mellitus; EV, enfortumab vedotin; GC, gemcitabine–cisplatin; gem–carbo, gemcitabine–carboplatin.

**Table 1 cancers-16-04078-t001:** Key factors involved in selecting the appropriate frontline therapy for patients with mUC.

Criteria	Category	Frontline Therapy Options
Cisplatin Eligibility [4,31,34]	Cisplatin-Eligible (eGFR ≥ 60 mL/min/1.73 m^2^)	- Gemcitabine + Cisplatin (standard of care) - Dose-dense MVAC (Methotrexate, Vinblastine, Doxorubicin, and Cisplatin) - Avelumab maintenance therapy after a response or stable disease following 4–6 cycles of chemotherapy.
	Cisplatin-Ineligible (eGFR < 60 mL/min/1.73 m^2^)	- Gemcitabine + Carboplatin (standard alternative) - Atezolizumab or Pembrolizumab (in patients with high PD-L1 expression or those who are not candidates for any platinum therapy) - Avelumab maintenance therapy following stable disease or response to chemotherapy.
PD-L1 Expression	High or intermediate (if platinum-Ineligible, Atezolizumab: SP142 assay, PD-L1–stained tumor-infiltrating immune cells covering ≥5% of the tumor area, Pembrolizumab: TPS ≥ 1%) [53,54]	- Atezolizumab or Pembrolizumab.
	Low or negative PD-L1 (Cisplatin-Ineligible)	- Gemcitabine + Carboplatin.
Performance Status (ECOG)	ECOG 0-1	- These patients typically tolerate platinum-based chemotherapy well, and cisplatin-based regimens are the standard. If cisplatin-ineligible, carboplatin-based regimens or PD-L1-targeted immunotherapy can be considered, especially in PD-L1 positive patients.
	ECOG 2	- For patients with moderate performance status (ECOG 2), carboplatin-based chemotherapy is preferred, as cisplatin may be too toxic. Immunotherapy is another option, especially in patients with high PD-L1 expression.
	ECOG ≥ 3	- Patients with poor performance status are generally not good candidates for chemotherapy. Atezolizumab or pembrolizumab monotherapy may be considered, especially in those with high PD-L1 expression. Supportive care or clinical trials are also options.
Other Factors	Comorbidities (e.g., cardiovascular disease)	- Patients with significant comorbidities that make cisplatin too toxic are typically treated with carboplatin-based chemotherapy or immunotherapy. Atezolizumab or pembrolizumab may be used for those who are cisplatin-ineligible and PD-L1 positive.
Molecular Profiling	FGFR2/3 alterations (approved criteria: Four-point mutations: R248C, S249C, G370C, Y373C, and Five fusions: FGFR2-BICC1, FGFR2-CASP7, FGFR3-TACC3 (variants V1 and V3), FGFR3-BAIAP2L1) [7,55]	- Patients with FGFR mutations who have progressed on platinum-based chemotherapy may be eligible for erdafitinib, an FGFR inhibitor, but this is not used in frontline therapy. However, molecular profiling can inform future treatment lines.

Abbreviation: TPS, tumor proportion score.

**Table 2 cancers-16-04078-t002:** Key clinical trials in frontline therapy for mUC.

Trial	Intervention Arm	Control Arm	Median PFS in the Intervention Arm (Months)	Median PFS in the Control Arm (Months)	Hazard Ratio (HR) (95% CI)	*p*-Value	Median OS in the Intervention Arm (Months)	Median OS in the Control Arm (Months)	Hazard Ratio (HR) (95% CI)	*p*-Value	Adverse Events
De Santis et al. 2009 [62]	Gemcitabine/Carboplatin	Methotrexate/Carboplatin/Vinblastine	Not available	Not available			9.3	8.1			
Sternberg et al. 2006 [60]	High-dose intensity M-VAC + G-CSF	Classic M-VAC	Not available	Not available			15.9	14.2		0.075	Reduced toxicity with dose-dense M-VAC
GC vs. MVAC [57]	Gemcitabine + Cisplatin	MVAC	7.0	7.5	Similar HR	0.8	14.0	15.2	Similar HR	0.6	GC better tolerability, lower toxicity (Grade 3+ AE)
KEYNOTE-901 [78]	Nivolumab + GC	Gemcitabine-cisplatin	7.9	7.6	0.78 (0.63–0.96)	0.02	21.7	18.9	0.78 (0.63–0.96)	0.02	Grade 3+ TRAEs 61.8% vs. 51.7%
JAVELIN-100 [82]	Avelumab (maintenance)	Best Supportive Care	5.5	2.1	0.69 (0.56–0.86)	0.001	21.4	14.3	0.69 (0.56–0.86)	0.001	Grade 3+ TRAEs 47% vs. 25%
DANUBE [63]	Durvalumab + tremelimumab	Chemotherapy	6.7	6.9	0.85 (0.71–1.02)	0.075	15.1	12.1	0.85 (0.72–1.02)	0.054	Grade 3+ TRAEs 61% vs. 50%
IMvigor130 [64]	Atezolizumab + chemo	Chemotherapy alone	8.2	6.3	0.82 (0.70–0.96)	0.007	16.0	13.4	0.83 (0.69–1.00)	0.027	Grade 3+ TRAEs 81% vs. 76%
IMvigor210 [70]	Atezolizumab (monotherapy)	No control (single-arm)	2.7	N/A	N/A	N/A	7.9	N/A	N/A	N/A	Grade 3–4 TRAEs 16%
EV-103 [94]	EV + Pembrolizumab	No control (single-arm)	12.3	N/A	N/A	N/A	26.1	N/A	N/A	N/A	Grade 3–4 TRAEs 54%
EV-302 [9]	EV + Pembrolizumab	Standard chemotherapy	12.5	6.3	0.45 (0.38–0.54)	<0.001	31.5	16.1	0.47 (0.38–0.58)	<0.001	Grade 3+ TRAEs 55.9% vs. 69.5%
KEYNOTE-052 [72]	Pembrolizumab (monotherapy)	No control (single-arm)	2.1	N/A	N/A	N/A	11.3	N/A	N/A	N/A	Grade 3–4 TRAEs 16%
KEYNOTE-361 [75]	Pembrolizumab + chemo	Chemotherapy alone	8.3	7.1	0.78 (0.65–0.93)	0.003	17.0	14.3	0.86 (0.72–1.02)	0.04	Grade 3–4 TRAEs 67% vs. 63%

Abbreviations: CI, Confidence Interval; GC, Gemcitabine+Cisplatin; Ev, Enfortumab Vedotin; G-CSF: Granulocyte-Colony Stimulating Factor; HR: Hazard Ratio; M-VAC: Methotrexate, Vinblastine, Doxorubicin, and Cisplatin; N/A: Not Available; OS: Overall Survival; PFS: Progression-Free Survival; TRAE: Treatment-Related Adverse Events.

**Table 3 cancers-16-04078-t003:** Recent ongoing first-line phase III/II clinical trials in patients with advanced UC.

Study	Cisplatin- Eligibility	Experimental Arm	Status	Phase	Enrolment (n)	Patients	Primary End Point
NCT03682068 (NILE)	cisplatin-eligible	Durvalumab + SoC (CT) or durvalumab + tremelimumab + SoC vs. SoC	Active, not recruiting	III	1292	La/mUC	OS
NCT03036098	cisplatin-eligible	Nivolumab + ipilimumab or + SoC CT vs. SoC	Active, not recruiting	III	1307	La/mUC	OS, PFS
NCT05302284	cisplatin-eligible	RC48-ADC + toripalimab vs. CT alone	Recruiting	III	452	La/mUC with HER2 expressing	OS, PFS
NCT03967977	cisplatin-eligible	Cis/Car + G + tislelizumab vs. Cis/Car + G + placebo	Recruiting	III	420	La/mUC	OS
NCT02567409	cisplatin-eligible	Cis + G ± M6620	Active, not recruiting	II	91 actuals	mUC	PFS
NCT04486781	cisplatin/platinum-ineligible	Pembrolizumab + sEphB4-HSA	Recruiting	II	38	mUC patients CT ineligible or refused	ORR
NCT03237780	cisplatin/platinum-ineligible	Eribulin mesylate + atezolizumab vs. atezolizumab alone	Active, not recruiting	II	72	La/mUC	AEs. ORR
NCT04601857	cisplatin/platinum-ineligible	Futibatinib + pembrolizumab	Active, not recruiting	II	46	La/mUC	ORR
NCT05645692	cisplatin/platinum-ineligible	RO7247669 ± tiragolumab vs. atezolizumab	Active, not recruiting	II	240	La/mUC	ORR

## Data Availability

No new data were created.

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
