# Peer review of "Evolving Treatment Landscape of Frontline Therapy for Metastatic Urothelial Carcinoma: Current Insights and Future Perspectives"

_cancers, 2024, doi:10.3390/cancers16234078_

Round 1
Reviewer 1 Report
Comments and Suggestions for Authors
This review provides an overview of the evolving frontline therapies for metastatic urothelial carcinoma, emphasizing the shift from traditional cisplatin-based chemotherapy to ICIs and antibody-drug conjugates (ADCs), notably the enfortumab vedotin and pembrolizumab combination. The author team summarized the most recent and relevent papers on the subject. Although there are several typos and the narrative is a kind of fragmentary, I just abide by scientific soundness. The text is well written and very easy to read and follow it. I would like to offer the following points for consideration by the authors towards the improvement of the manuscript:
1- The review mentions ICIs as a treatment option, but there is no detailed analysis of how patients with different PD-L1 expression levels respond to ICIs.
2- While the review discusses various histological subtypes, it does not cover specific outcomes or benefits for patients with sarcomatoid differentiation.
3- The article mainly references clinical trial data, but it may be useful to include up-to-date real-world evidence reflecting treatment outcomes in broader patient populations.
Comments on the Quality of English LanguageThe English could be improved to more clearly express the research.
Author Response
Comments and Suggestions for Authors
This review provides an overview of the evolving frontline therapies for metastatic urothelial carcinoma, emphasizing the shift from traditional cisplatin-based chemotherapy to ICIs and antibody-drug conjugates (ADCs), notably the enfortumab vedotin and pembrolizumab combination. The author team summarized the most recent and relevent papers on the subject. Although there are several typos and the narrative is a kind of fragmentary, I just abide by scientific soundness. The text is well written and very easy to read and follow it. I would like to offer the following points for consideration by the authors towards the improvement of the manuscript:
1- The review mentions ICIs as a treatment option, but there is no detailed analysis of how patients with different PD-L1 expression levels respond to ICIs.
Response: Thank you for your thoughtful suggestion. We added sentences about how patients with different PD-L1 expression levels respond to ICIs as follows.
Revisions:
Page 6, line 250-252
4.2.1 The revised ORRs by PD-L1 subgroup were 28% (95% confidence interval [CI]: 14–47) in IC2/3, 24% (95% CI: 15–35) in IC1/2/3, 21% (95% CI: 10–35) in IC1, and 21% (95% CI: 9–36) in IC0.
Page 6, line 269-272
In patients with a PD-L1 combined positive score (CPS) ≥10, the ORR was 47.3%, with 20.0% patients achieving CR; for those with a PD-L1 CPS <10, the ORR was 20.3%. The mOS was 18.5 months (95% CI, 12.2–28.5 months) in the CPS ≥10 group and 9.7 months (95% CI, 7.6–11.5 months) in the CPS <10 group.
Page 7, line 320-323
4.2.2 Among patients with tumor PD-L1 expression of 1% or higher, the nivolumab combination demonstrated favorable HR compared to gemcitabine–cisplatin alone. For overall survival, the HR was 0.75 (95% CI, 0.53–1.06), and for PFS as assessed by central review, the HR was 0.60 (95% CI, 0.41–0.81).
Page 8, line 350-355
4.2.3 In the PD-L1–positive population, the avelumab group showed significantly longer OS compared to the control group, with a 1-year survival rate of 79.1% (95% CI, 72.1–84.5) versus 60.4% (95% CI, 52.0–67.7) in the control group (stratified HR, 0.56; 95% CI, 0.40–0.79; P<0.001). Among patients with PD-L1–negative tumors, the mOS was 18.8 months (95% CI, 13.3–22.5) for the avelumab group compared to 13.7 months (95% CI, 10.8–17.8) in the control group (stratified HR, 0.85; 95% CI, 0.62–1.18).
Page 11, line 542-545
- A recent meta-analysis of 14 studies showed that PD-L1-positive UC is associated with higher response rates and improved survival outcomes in patients treated with ICIs. Although PD-L1 positivity suggests a better prognosis, it is not a reliable predictive biomarker for ICI response 40.
2- While the review discusses various histological subtypes, it does not cover specific outcomes or benefits for patients with sarcomatoid differentiation.
Response: Thank you for the valuable suggestion. We have described specific outcomes or benefits for patients with various histological subtypes.
Revisions:
Page 4-5, line 191-199
Data on ICIs and new therapies for micropapillary, sarcomatoid, and neuroendocrine UC are limited. ICIs showed a 28% overall response rate (ORR) in 25 patients, and EV achieved a 35% ORR in 41 patients with micropapillary components 42, 43. Advanced sarcomatoid UC exhibits higher PD-L1 expression with 35–40% response rates to ICIs 42; however, low nectin-4 expres-sion may limit EV and pembrolizumab efficacy 44. For neuroendocrine UC, single-agent an-ti-PD-1/PD-L1 inhibitors, as well as lurbinectedin and sacituzumab govitecan have demonstrat-ed some efficacy in small studies 45-48, but the effectiveness of dual checkpoint blockade is un-certain 49, 50, and low nectin-4 expression may limit EV’s benefits 43, 44.
3- The article mainly references clinical trial data, but it may be useful to include up-to-date real-world evidence reflecting treatment outcomes in broader patient populations.
Response: Thank you for the valuable suggestion. We have added section 4.4 about up-to-date real-world evidence reflecting treatment outcomes and patterns in broader patient populations.
Revisions:
Page 10, line 449-487
4.4 Real-world 1L treatment patterns and survival rates for la/mUC
Several retrospective studies have analyzed real-world 1L treatment patterns and OS in patients with la/mUC in the United States and Germany 92-94. These regions were chosen owing to their high incidence of UC and availability of comprehensive healthcare data that support real-world analysis.
A large-scale analysis of Medicare fee-for-service claims from 2015–2022 involving 13,104 U.S. patients with UC revealed that PBC remains the most commonly administered 1L treatment. Specifically, 19.7% patients received cisplatin and 28.4% received carboplatin. ICI monotherapy was also prevalent, being administered to 34.8% of patients, particularly those with comorbidi-ties such as renal disease (39.9%), chronic obstructive pulmonary disease (36.1%), and diabetes (32.2%). Despite cisplatin-treated patients showing the longest median OS at 17.0 months, over-all outcomes were generally poor across all treatment groups, largely due to disease progres-sion, limited efficacy of available treatments, and patient comorbidities that impacted tolerabil-ity and response 94.
Following the FDA approval of avelumab maintenance therapy in June 2020, a study utilized the U.S. Oncology Network electronic health records from December 2019 to September 2022 and assessed 1,072 patients with la/mUC (median age: 73 years, 74.9% male). 1L treatments in-cluded ICI monotherapy (43.8%), PBC alone (37.3%), and PBC followed by avelumab mainte-nance (10.4%). Notably, ICI monotherapy was frequently used even in patients eligible for PBC. Patients receiving PBC with subsequent avelumab maintenance exhibited the longest median OS (20 months for cisplatin plus avelumab and 18.4 months for carboplatin plus avelumab), supporting avelumab maintenance as a standard option for those without disease progression on PBC. Thus, incorporating maintenance therapy into clinical practice may potentially extend patient survival, especially in those responding well to initial PBC 93.
In Germany, the CONVINCE retrospective multicenter study evaluated 188 patients treated between 2019 and 2021 (median age: 70 years, 72.3% male). The majority (76.1%) received PBC, predominantly combined with gemcitabine, while 19.1% received ICI monotherapy, mainly atezolizumab or pembrolizumab. A small group received other non-PBC treatments. The study found high adherence to treatment guidelines recommending PBC as the 1L therapy. The emerging use of ICI therapies, including avelumab maintenance, indicates potential for im-proved patient outcomes and necessity for further real-world research to assess their integration into clinical practice 92.
Collectively, these studies demonstrate that while PBC remains the standard 1L treatment asso-ciated with longer OS in patients with la/mUC, the OS rates are suboptimal, primarily due to disease biology, treatment tolerability, and patient-specific factors such as comorbidities. The increasing adoption of ICI therapies and maintenance treatments such as avelumab may en-hance outcomes. However, the substantial use of ICI monotherapy, even among patients who could tolerate PBC, suggests opportunities to optimize treatment strategies. These findings un-derscore an unmet need for more effective 1L therapies to improve survival of patients with la/mUC.
Comments on the Quality of English Language
The English could be improved to more clearly express the research.
Response: Thank you for your comment. Although the manuscript was previously proofread in English, upon review, we recognize that further improvements can be made to more clearly express the research. We apologize for any shortcomings in the language and have made comprehensive revisions to the manuscript, including re-editing the entire document for clarity and accuracy.
Revisions:
Page 13, line 613-614
Acknowledgement: We would like to thank Editage (www.editage.co.kr) for English language editing.

Reviewer 2 Report
Comments and Suggestions for Authors
The authors provide a timely, well organised review describing the current first-line treatment strategy for metastatic urothelial carcinoma (mUC), relying on chemotherapy as well as advanced immunotherapies comprising immune checkpoint inhibitors (ICIs) and antibody-drug conjugates (ADCs). As this review offers an excellent overview of the current status and future prospects in the oncology of mUC, it is of interest to the readership of Cancers.
Some amendments are recommended:
1. Lung toxicity may be included as an additional AE of Enfortumab Vedotin treatment, as significant anecdotical evidence was found in the recently published paper of Desimpel et al. Lung Toxicity Occurring During Enfortumab Vedotin Treatment: From a Priming Case Report to a Retrospective Analysis. Pharmaceuticals 2024, 17, 1547. https://doi.org/10.3390/ph17111547.
2. Though having a substantial commercial impact, it is clear that the therapeutic success of the standard cancer therapies (including ICIs and ADCs) is insufficient as can be deduced from the increasing number of patients, severe AEs requiring further treatments, etc., thus an actual cure can hardly be offered. These problems are addressed by the authors, but essentially only within the framework of chemotherapy, ICIs and AEs.
Nonetheless, alternative low-risk measures as combined therapies should be mentioned, the more so as these strengthen the immune system: For example, plant nutrients as part of the diet, such as ginger derivatives are of substantial interest providing direct anti-tumour effects as well as being chemopreventive, thus mitigating AEs from ICI, etc. (for review: Zadorozhna and Mangieri. Mechanisms of Chemopreventive and Therapeutic Proprieties of Ginger Extracts in Cancer. Int. J. Mol. Sci. 2021, 22, 6599. https://doi.org/10.3390/ijms22126599).
Similarly, high doses of properly administered ascorbic acid are of interest (E.g. for review: Isola et al. Vitamin C Supplementation in the Treatment of Autoimmune and Onco-Hematological Diseases: From Prophylaxis to Adjuvant Therapy. Int. J. Mol. Sci. 2024, 25, 7284. https://doi.org/10.3390/ijms25137284). In fact, there is a tremendous potential of high dose ascorbic acid treatment as indicated in a large body of literature for many decades, which is still largely ignored in the current health system and academia (for classical reviews: Klenner. Observations on the dose and administration of ascorbic acid when employed beyond the range of a vitamin in human pathology. J. Appl. Nutr, 1971; 23(3&4): 60-89. Klenner. Significance of high daily intake of ascorbic acid in preventive medicine. J. Int. Acad. Prev. Med, 1974; 1:1, 45-69.).
Furthermore, simple but effective measures addressing metabolic peculiarities of tumor cells, e.g. fasting and strict avoidance of sugar uptake (i.e. cutting off tumor cells from glucose as an essential nutrient) or increasing the pH level in the TME (as tumors can only grow in an acidic environment) are of interest.
Author Response
Comments and Suggestions for Authors
The authors provide a timely, well organised review describing the current first-line treatment strategy for metastatic urothelial carcinoma (mUC), relying on chemotherapy as well as advanced immunotherapies comprising immune checkpoint inhibitors (ICIs) and antibody-drug conjugates (ADCs). As this review offers an excellent overview of the current status and future prospects in the oncology of mUC, it is of interest to the readership of Cancers.
Some amendments are recommended:
- Lung toxicity may be included as an additional AE of Enfortumab Vedotin treatment, as significant anecdotical evidence was found in the recently published paper of Desimpel et al. Lung Toxicity Occurring During Enfortumab Vedotin Treatment: From a Priming Case Report to a Retrospective Analysis. Pharmaceuticals 2024, 17, 1547. https://doi.org/10.3390/ph17111547.
Response: Thank you for the valuable suggestion. We have added sentence about lung toxicity and cited papers on the subject based on your comment.
Revisions:
Page 12, line 554-555
5.3 Additionally, a recently reported promising case of lung toxicity also emphasizes the need for close pulmonary monitoring during EV treatment 106.
- Though having a substantial commercial impact, it is clear that the therapeutic success of the standard cancer therapies (including ICIs and ADCs) is insufficient as can be deduced from the increasing number of patients, severe AEs requiring further treatments, etc., thus an actual cure can hardly be offered. These problems are addressed by the authors, but essentially only within the framework of chemotherapy, ICIs and AEs.
Nonetheless, alternative low-risk measures as combined therapies should be mentioned, the more so as these strengthen the immune system: For example, plant nutrients as part of the diet, such as ginger derivatives are of substantial interest providing direct anti-tumour effects as well as being chemopreventive, thus mitigating AEs from ICI, etc. (for review: Zadorozhna and Mangieri. Mechanisms of Chemopreventive and Therapeutic Proprieties of Ginger Extracts in Cancer. Int. J. Mol. Sci. 2021, 22, 6599. https://doi.org/10.3390/ijms22126599).
Similarly, high doses of properly administered ascorbic acid are of interest (E.g. for review: Isola et al. Vitamin C Supplementation in the Treatment of Autoimmune and Onco-Hematological Diseases: From Prophylaxis to Adjuvant Therapy. Int. J. Mol. Sci. 2024, 25, 7284. https://doi.org/10.3390/ijms25137284). In fact, there is a tremendous potential of high dose ascorbic acid treatment as indicated in a large body of literature for many decades, which is still largely ignored in the current health system and academia (for classical reviews: Klenner. Observations on the dose and administration of ascorbic acid when employed beyond the range of a vitamin in human pathology. J. Appl. Nutr, 1971; 23(3&4): 60-89. Klenner. Significance of high daily intake of ascorbic acid in preventive medicine. J. Int. Acad. Prev. Med, 1974; 1:1, 45-69.).
Furthermore, simple but effective measures addressing metabolic peculiarities of tumor cells, e.g. fasting and strict avoidance of sugar uptake (i.e. cutting off tumor cells from glucose as an essential nutrient) or increasing the pH level in the TME (as tumors can only grow in an acidic environment) are of interest.
Response: Thank you for the valuable suggestion. We have added section 4.4 based on your suggestion (Zadorozhna and Mangieri. Mechanisms of Chemopreventive and Therapeutic Proprieties of Ginger Extracts in Cancer. Int. J. Mol. Sci. 2021, 22, 6599. https://doi.org/10.3390/ijms22126599, Isola et al. Vitamin C Supplementation in the Treatment of Autoimmune and Onco-Hematological Diseases: From Prophylaxis to Adjuvant Therapy. Int. J. Mol. Sci. 2024, 25, 7284. https://doi.org/10.3390/ijms25137284, Klenner. Observations on the dose and administration of ascorbic acid when employed beyond the range of a vitamin in human pathology. J. Appl. Nutr, 1971; 23(3&4): 60-89. Klenner. Significance of high daily intake of ascorbic acid in preventive medicine. J. Int. Acad. Prev. Med, 1974; 1:1, 45-69.)
Revisions:
Page 12, line 561-569
5.4 Combining low-risk dietary and metabolic therapies with standard cancer treatments to en-hance efficacy and mitigate toxicity
The therapeutic effectiveness of standard cancer therapies, such as ICIs and ADCs, is limited by severe AEs. Therefore, there is a need for low-risk combined therapies that enhance the immune system. Plant-based dietary nutrients, such as ginger derivatives, show anti-tumor effects and can mitigate ICI-related AEs 109. High-dose ascorbic acid has significant potential, but is un-derutilized 110-112. Metabolic interventions such as fasting, avoiding sugar to starve tumor cells, and increasing tumor pH to inhibit growth have been considered 113. Combining these measures with standard treatments has the potential to improve efficacy and reduce toxicity.

Reviewer 3 Report
Comments and Suggestions for Authors
Manuscript entitled "Evolving treatment landscape of frontline therapy for metastatic urothelial carcinoma: Current Insights and Future Perspectives"
This work is of interest while some modifications should be made:
1. The authors should describe more recent clinical trials and prepare a table summarzing these trials.
2. The authors should have more discussion on the recently approved treatments, including but not limited to Nectin, etc.
3. In the table summarizing key factors, the authors should list the criteria. For example, for PD-L1, the antibody clones and score (TC, CPS, ... etc.) and the definition of positivity should be mentioned. The mutation patterns for FGFR2/3 should also be mentioned.
4. The authors should summarize the survival benefits of the different therapeutic protocols.
Author Response
Comments and Suggestions for Authors
Manuscript entitled "Evolving treatment landscape of frontline therapy for metastatic urothelial carcinoma: Current Insights and Future Perspectives"
This work is of interest while some modifications should be made:
- The authors should describe more recent clinical trials and prepare a table summarzing these trials.
Response: Thank you for the valuable suggestion. We have added sentences and Table 3 (updated in detail) based on your suggestion
Revisions:
Page 11, line 508-510
The evolving landscape of frontline therapy for mUC presents significant advancements, with many ongoing Phase III and II clinical trials actively contributing to the refinement of future treatment paradigms (Table 3).
Table 3. Recent ongoing First line phase III/II clinical trials in patients with advanced UC
|
Study |
cisplatin-eligiblility |
Experimental arm |
Status |
Phase |
Enrolment (n) |
Patients |
Primary end point |
|
NCT03682068 (NILE) |
cisplatin-eligible |
Durvalumab + SoC (CT) or durvalumab + tremelimumab + SoC vs SoC |
Active, not recruiting |
III |
1292 |
La/mUC |
OS |
|
NCT03036098 |
cisplatin-eligible |
Nivolumab + ipilimumab or + SoC CT vs SoC |
Active, not recruiting |
III |
1307 |
La/mUC |
OS, PFS |
|
NCT05302284 |
cisplatin-eligible |
RC48-ADC + toripalimab vs CT alone |
Recruiting |
III |
452 |
La/mUC with HER2 expressing |
OS, PFS |
|
NCT03967977 |
cisplatin-eligible |
Cis/Car + G + tislelizumab vs Cis/Car + G + placebo |
Recruiting |
III |
420 |
La/mUC |
OS |
|
NCT02567409 |
cisplatin-eligible |
Cis + G ± M6620 |
Active, not recruiting |
II |
91 actuals |
mUC |
PFS |
|
NCT04486781 |
cisplatin/ platinum-ineligible |
Pembrolizumab + sEphB4-HSA |
Recruiting |
II |
38 |
mUC patients CT ineligible or refused |
ORR |
|
NCT03237780 |
cisplatin/ platinum-ineligible |
Eribulin mesylate + atezolizumab vs atezolizumab alone |
Active, not recruiting |
II |
72 |
La/mUC |
AEs. ORR |
|
NCT04601857 |
cisplatin/ platinum-ineligible |
Futibatinib + pembrolizumab |
Active, not recruiting |
II |
46 |
La/mUC |
ORR |
|
NCT05645692 |
cisplatin/ platinum-ineligible |
RO7247669 ± tiragolumab vs atezolizumab |
Active, not recruiting |
II |
240 |
La/mUC |
ORR |
- The authors should have more discussion on the recently approved treatments, including but not limited to Nectin, etc.
Response: Thank you for your thoughtful suggestion. We have added section 5.5 based on your suggestion
Revisions:
Page 12, line 570-585
5.5 Clinical significance of newer treatments other than EV-302
The currently recommended treatment algorithm (Figure 3) for mUC outlines strategies based on cisplatin eligibility and patient-specific factors, and incorporates recent evidences to guide therapeutic decisions. For patients with visceral and/or bone metastasis, upper tract UC, high-risk cardiac disease, or good performance status, regardless of cisplatin eligibility, EV/pembro is recommended because of improved survival outcomes. In cisplatin-eligible pa-tients with comorbidities such as poorly controlled diabetes or retinal/corneal abnormalities, GC/nivo followed by maintenance nivolumab or GC followed by maintenance avelumab is advised, enabling a tailored approach based on disease response and tolerability. For cispla-tin-ineligible patients with peripheral neuropathy (Grade ≥2) or other vulnerabilities, gem-citabine-carboplatin followed by maintenance avelumab remains the standard of care. Finally, for patients who are platinum-ineligible, pembrolizumab monotherapy is the primary option, particularly for those who cannot tolerate platinum-based regimens. This algorithm reflects a personalized approach, optimizing the use of combination therapies such as EV/pembro and maintenance strategies with avelumab or nivolumab, while addressing the unmet needs of pa-tients with significant comorbidities or platinum ineligibility.
- In the table summarizing key factors, the authors should list the criteria. For example, for PD-L1, the antibody clones and score (TC, CPS, ... etc.) and the definition of positivity should be mentioned. The mutation patterns for FGFR2/3 should also be mentioned.
Response: Thank you for the valuable suggestion. We have added the criteria about PD-L1 and FGFR2/3 alterations
Revisions:
Table 1. key factors involved in selecting the appropriate frontline therapy for patients with mUC.
|
Criteria |
Category |
Frontline therapy Options |
|
Cisplatin Eligibility 4, 31, 34 |
Cisplatin-Eligible (eGFR ≥ 60 mL/min/1.73 m²) |
- Gemcitabine + Cisplatin (standard of care) |
|
|
Cisplatin-Ineligible (eGFR < 60 mL/ min/1.73 m²) |
- Gemcitabine + Carboplatin (standard alternative) |
|
PD-L1 Expression |
High or intermediate (if platinum-Ineligible, Atezolizumab: SP142 assay, PD-L1–stained tumor-infiltrating immune cells covering ≥5% of the tumor area, Pembrolizumab: TPS ≥1% ) 65, 114 |
- Atezolizumab or Pembrolizumab. |
|
|
Low or negative PD-L1 (Cisplatin-Ineligible) |
- Gemcitabine + Carboplatin. |
|
Performance Status (ECOG) |
ECOG 0-1 |
- These patients typically tolerate platinum-based chemotherapy well, and cisplatin-based regimens are the standard. If cisplatin-ineligible, carboplatin-based regimens or PD-L1-targeted immunotherapy can be considered, especially in PD-L1 positive patients. |
|
|
ECOG 2 |
- For patients with moderate performance status (ECOG 2), carboplatin-based chemotherapy is preferred, as cisplatin may be too toxic. Immunotherapy is another option, especially in patients with high PD-L1 expression. |
|
|
ECOG ≥ 3 |
- Patients with poor performance status are generally not good candidates for chemotherapy. Atezolizumab or pembrolizumab monotherapy may be considered, especially in those with high PD-L1 expression. Supportive care or clinical trials are also options. |
|
Other Factors |
Comorbidities (e.g., cardiovascular disease) |
- Patients with significant comorbidities that make cisplatin too toxic are typically treated with carboplatin-based chemotherapy or immunotherapy. Atezolizumab or pembrolizumab may be used for those who are cisplatin-ineligible and PD-L1 positive. |
|
Molecular Profiling |
FGFR2/3 alterations (approved criteria: Four-point mutations: R248C, S249C, G370C, Y373C, and Five fusions: FGFR2-BICC1, FGFR2-CASP7, FGFR3-TACC3 (variants V1 and V3), FGFR3-BAIAP2L1) 7, 115 |
- Patients with FGFR mutations who have progressed on platinum-based chemotherapy may be eligible for erdafitinib, an FGFR inhibitor, but this is not used in frontline therapy. However, molecular profiling can inform future treatment lines. |
- The authors should summarize the survival benefits of the different therapeutic protocols.
Response: Thank you for your thoughtful suggestions. We have added section 4.5 based on your suggestion
Revisions:
Page 10-11, line 488-506
4.5 Survival benefits of the different therapeutic protocols
Among cisplatin-eligible patients, gemcitabine plus cisplatin, the historical standard, achieved a mOS of 13.8 months 54. The addition of nivolumab (CheckMate 901) improved the OS to 21.7 months 76. EV plus pembrolizumab (EV-302) reported the longest OS at 31.5 months 9.
For cisplatin-ineligible patients, gemcitabine plus carboplatin resulted in a median OS of 9.3 months 95, highlighting the limitations of carboplatin-based chemotherapy, which is generally less effective owing to its lower potency and reduced ability to form DNA cross-links compared to cisplatin. Pembrolizumab monotherapy (KEYNOTE-052), approved for platinum-ineligible patients, demonstrated an OS of 11.3 months but remains less effective than combination thera-pies 70.
Avelumab maintenance therapy (JAVELIN 100), following disease control with PBC, signifi-cantly prolonged survival 62. The median OS was 25.8 months for patients receiving carboplatin during induction and 31.5 months for those receiving cisplatin. This highlights the specific role of avelumab in maintaining disease control and extending OS, particularly in patients who ini-tially respond to platinum-based induction therapy.
However, outcomes for cisplatin-ineligible patients remain suboptimal, as reflected by the poorer OS with carboplatin-based or immunotherapy monotherapies, emphasizing the need for further research, including the development of novel targeted therapies and improved ADCs, and exploring combination regimens with better response rates.

Round 2
Reviewer 3 Report
Comments and Suggestions for Authors
The revision is acceptable for publication.